# Biases in ice sheet models from missing noise-induced drift

Alexander A. Robel[1,*], Vincent Verjans[1,2,*], and Aminat A. Ambelorun[1]

[1]School of Earth and Atmospheric Sciences, Georgia Institute of Technology, Atlanta, GA, USA
[2]Center for Climate Physics, Institute for Basic Science, Busan, Republic of Korea
[*]Both authors equally contributed to this study

**Correspondence:** Alexander A. Robel (robel@eas.gatech.edu)

**Abstract.** Most climatic and glaciological processes exhibit internal variability, which is omitted from many ice sheet model simulations. Prior studies have found that climatic variability can change ice sheet sensitivity to the long-term mean and trend in climate forcing. In this study, we use an ensemble of simulations with a stochastic large-scale ice sheet model to demonstrate that variability in frontal ablation of marine-terminating glaciers changes the mean state of the Greenland Ice Sheet through noise-induced drift. Conversely, stochastic variability in surface mass balance does not appear to cause noise-induced drift in these ensembles. We describe three potential causes for noise-induced drift identified in prior statistical physics literature: noise-induced bifurcations, multiplicative noise, and nonlinearities in noisy processes. Idealized simulations and Reynolds decomposition theory show that for marine ice sheets in particular, noise-induced bifurcations and nonlinearities in variable ice sheet processes are likely the cause of the noise-induced drift. We argue that the omnipresence of variability in climate and ice sheet systems means that the state of real-world ice sheets includes this tendency to drift. Thus, the lack of representation of such noise-induced drift in spin-up and transient ice sheet simulations is a potentially ubiquitous source of bias in ice sheet models.

## 1  Introduction

The Earth system exhibits internal variability in many processes on a wide range of time scales. As one component of the Earth system, ice sheets are subject to variability in climatic processes, including snowfall, atmospheric temperatures, and ocean currents. Ice sheets also exhibit internal variability of their own, in processes related to hydrology, ice fracture and ice flow. In general, numerical ice sheet modeling studies focus on the ice sheet response to changes in the mean forcing, often without including internal variability in climate or glaciological systems (e.g., Golledge et al., 2015; DeConto et al., 2021). The central assumption of such studies is that the long-term state of glaciers and ice sheets is set only by the multi-decadal mean and trend in climate forcing. This assumption is based on the long equilibrium timescale of glaciers and ice sheets (Nye, 1960; Oerlemans and Van Der Veen, 1984; Robel et al., 2018). However, critically, this long response time scale does not imply that glaciers and ice sheets are insensitive to short-timescale climatic fluctuations (Roe and O'Neal, 2009). Several recent studies, most using idealized glacier and ice sheet models, have demonstrated that this assumption may not hold in many circumstances known to exist in the real world. In land-based ice sheets with stochastic variability of surface temperature (Mikkelsen et al., 2018; Lauritzen et al., 2023) or marine-based ice sheets with periodic variability in ice viscosity (Hindmarsh and Le Meur,

2001), stochastic variability of ice shelf length (Robel et al., 2018), or seasonal variability of the calving front (Felikson et al., 2022), the inclusion of variability causes drift of the ice sheet state. This phenomenon of "noise-induced drift" is well known in the statistical physics community, where many useful mathematical tools have been developed to understand the cause of this phenomenon (e.g., Kloeden and Platen, 1995; Horsthemke and Lefever, 1984).

In this study, we show that noise-induced drift in response to stochastic frontal ablation is expected to occur in real marine ice sheets, and numerical modeling of marine ice sheets. This is demonstrated with ensemble simulations of the Greenland ice sheet, resembling modern conditions with realistic stochastic variability in frontal ablation. Ensembles with stochastic forcing in surface mass balance do not exhibit the same noise-induced drift, though other studies using stochastic surface temperature forcing in parameterized surface mass balance schemes do exhibit such drift (Mikkelsen et al., 2018). We describe the three

different potential mechanisms of noise-induced drift in generic stochastic systems, and identify which of these mechanisms are likely to cause noise-induced drift in real ice sheets. We close by arguing that modern ice sheet models omitting variability in climate and glaciological processes could produce biased estimates of the ice sheet mean state and the ice sheet response to climate change. We provide two potential solutions for this problem in the initialization and forcing of ice sheet models.

## 2   The Greenland Ice Sheet Under Variable Forcing

The central goal of this study is to demonstrate that the response of ice sheets to long-term (decadal -millennial) climatic forcing depends on the inclusion and magnitude of variability in climate and glaciological processes. To achieve this goal, we run four ensembles of Greenland ice sheet simulations using the Stochastic Ice Sheet and Sea Level System Model (StISSM; Verjans et al., 2022). The core of this model is ISSM, which solves for the ice thickness and velocity on a finite-element mesh refined in locations of interest (Larour et al., 2012). In this study, we use the Shallow Shelf Approximation (SSA; MacAyeal,

1989) and refine the mesh at 11 large marine-terminating glacier catchments where the ice sheet margin evolves dynamically. All simulations are initialized at a deterministic steady-state. This configuration is meant to resemble the modern state of the Greenland ice sheet, but deviates somewhat from the real ice sheet which is not at a steady-state (Otosaka et al., 2022). This initial deterministic steady-state comes from a long spin-up run over 31,000 years with temporally constant forcing in surface mass balance (SMB), and ablation at calving fronts (described in more detail in Verjans et al., 2022). SMB at model mesh

elements is set according to an elevation-dependent profile which is fit separately in 19 catchments encompassing the entire ice sheet (Zwally et al., 2012), to resemble mean 1961-1990 SMB simulated in RACMO2 (Ultee et al., 2024; Ettema et al., 2009). Each marine-terminating catchment has a prescribed rate of ocean melt at calving fronts, based on thermal forcing from Wood et al. (2021). In the spinup, calving rates at each catchment are calibrated to produce a steady-state ice sheet configuration resembling the present-day ice sheet. We apply the Budd sliding law (Budd et al., 1979):

$$\boldsymbol{\tau_b} = -C^2 \boldsymbol{u_b} N, \tag{1}$$

where $\boldsymbol{\tau_b}$ is the basal friction, $\boldsymbol{u_b}$ is the basal sliding speed, and $C^2$ is a space-varying coefficient. Effective pressure, $N$, is set to maintain local hydrostatic equilibrium with the ocean throughout the ice-covered model domain (Tsai et al., 2015)

$$N = \rho_i gh + \rho_w gb, \tag{2}$$

where $\rho_{i,w}$ are the densities of ice and water, respectively, $g$ is the acceleration due to gravity, $h$ is the ice thickness and $b$ is the bed elevation. Initialized from this steady-state, a deterministic control run with temporally constant forcings, exhibits an increase in ice mass by only 0.07% in 2000 years. The spatial pattern of ice thickness change in this deterministic control run (not plotted) shows weak thickness changes which are uniformly distributed over catchments, indicating no significant changes to glacier termini.

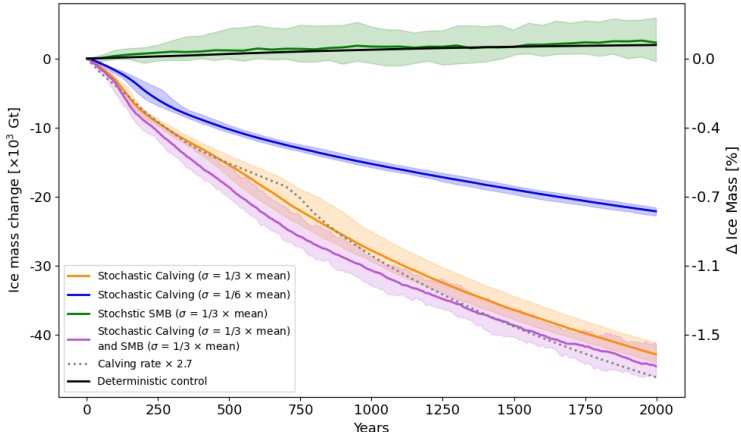

**Figure 1.** Ensemble mean and range of ice sheet mass change over four stochastic ensembles simulating the Greenland Ice Sheet. Yellow line and shading: white noise variability in calving rate with standard deviation 1/3 of mean. Blue line and shading: white noise variability in calving rate with standard deviation 1/6 of mean. Green line and shading: white noise variability in SMB with standard deviation 1/3 of mean. Purple line and shading: white noise variability in both calving rate and SMB, each with standard deviation 1/3 of mean. Shadings show the entire 10-member range. Black line is deterministic (i.e. no variability in forcing) simulation. Black dashed line is deterministic but with calving rates multiplied by 2.7.

We run ensembles of ten member simulations each, applying stochastic variability separately in SMB and calving rate, and we quantify the role of each forcing in setting the ice sheet state. Realistic stochastic parameterizations for SMB and ocean thermal forcing (which determines frontal melt) have been described in previous studies (Ultee et al., 2024; Verjans et al., 2023). These studies have found that variability in both SMB and ocean thermal forcing around Greenland is best described by autoregressive moving-average models of low order. In this study, for ease of interpretability, we conservatively apply simple white noise to different forcing variables with mean that remains constant in time and equal to deterministic steady-state values. White noise is characterized by independent random perturbations drawn from a Gaussian distribution, and with no autocorrelation in time. For both stochastic ensembles, the standard deviation of the stochastic variable in each catchment

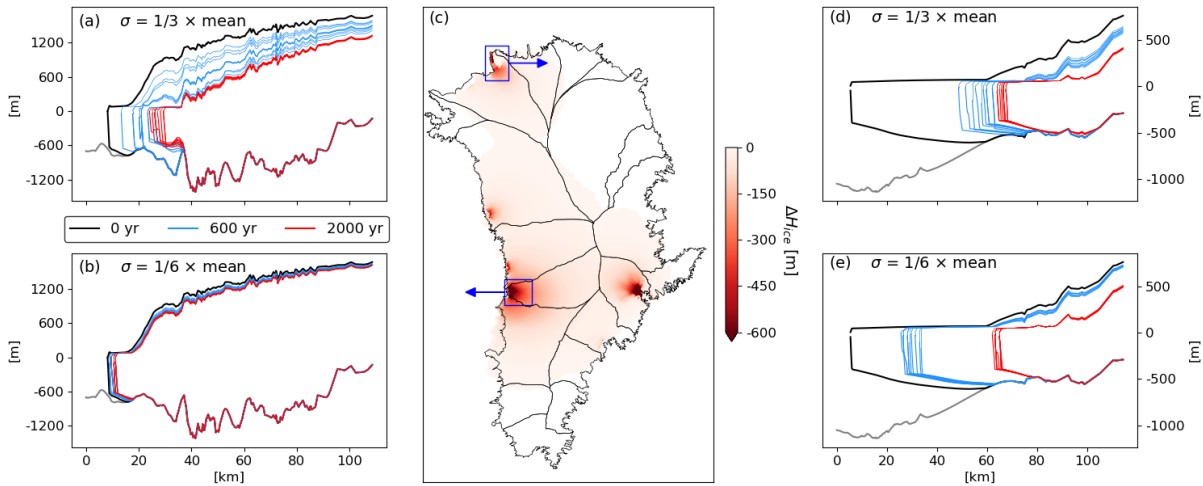

**Figure 2.** Ice thickness change for stochastic calving ensembles. (a) Profiles of ice thickness for all ensemble members at Sermeq Kujalleq (also called Jakobshavn Isbrae) for high amplitude variability in calving rate. Black line is initial glacier state for all simulations, blue lines are ensemble members after 600 years and red lines are after 2000 years. (b) Same as (a) but for lower amplitude variability in calving rate. (c) Ensemble mean ice thickness change for all of Greenland. (d-e) Same as (a-b) but for Petermann Glacier. Catchment delineations (Zwally et al., 2012) are shown in (c).

is set to 1/3 of the mean in that catchment. This amplitude of variability is chosen for simplicity but is similar to variability from observations and high-fidelity models of SMB and ocean forcing. In particular, Fettweis et al. (2020) finds that averaged across 13 different SMB models calibrated against observations, Greenland-wide SMB has a temporal standard deviation which is approximately 40% of the mean. Hanna et al. (2011) develop observation-based reanalyses of Greenland SMB over the 20th century, which also indicate a temporal standard deviation which is approximately 25-35% of the mean (depending on calibration dataset used). Verjans et al. (2023) finds that interannual variability of thermal forcing (which drives frontal ablation at glaciers) in the Estimating the Circulation and Climate of the Ocean Arctic Reanalysis product (ECCO; Nguyen et al., 2012) typically ranges between 10-60% around Greenland. As a point of comparison, we also run a fourth ensemble with the standard deviation of the stochastic calving rate equal to a conservatively low 1/6 of the mean calving rate.

In implementing white noise forcing in SMB and frontal ablation rate, we introduce symmetric variability directly in terms of the mass conservation equations for the ice sheet. This simplifies the task of identifying potential causes of resulting noise-induced drift, since the only dynamics to consider are those related to ice sheet flow. However, it may be that in reality, symmetric variability occurs in variables more removed from ice sheet dynamics such as atmospheric or ocean temperatures. Then, asymmetries or nonlinearities in the dependence of mass fluxes on these variables can be an additional source of noise-induced drift, as previously discussed by Mikkelsen et al. (2018) and Lauritzen et al. (2023). Our goal in this study is to identify mechanisms of noise-induced drift that are inherent in the fundamental dynamics of ice sheet flow. Such mechanisms would

be common to all ice sheet models, and not dependent on the model-specific parameterizations of mass fluxes as a function of climate forcing.

Ensemble simulations are run for 2000 years in order to observe the ice sheet evolution towards a new state. However, we note that an ice sheet the size of Greenland likely requires more than 10,000 years to reach a new steady-state in response to an ice-sheet-wide change in forcing due to long-term dynamic adjustment extending through the interior. Such long simulations are computationally challenging to perform for the entire Greenland Ice Sheet on a well-resolved mesh. The design of this ensemble was initially inspired by the larger Greenland ice sheet ensemble used to benchmark StISSM in Verjans et al. (2022),
which showed that just 10 ensemble members are sufficient to constrain the ensemble mean ice sheet mass to less than 0.1% of the converged values (albeit under different stochastic forcing). We also note here that in this depth-averaged model, the dynamic influence of calving and ocean melt at glacier termini is identical. We have chosen to implement stochastic calving in this study, but the results would be identical if stochastic frontal melt were implemented instead.

      Figure 1 shows the evolution of Greenland ice sheet mass over time from these ensemble simulations (colored lines and
shading) in comparison to the deterministic control simulation (black line). The most striking result is that stochastic variability in calving at marine-terminating glaciers causes substantial drift in the ensemble-mean ice sheet mass (yellow and blue lines). This drift is apparent in all ensemble members and exceeds the spread of intra-ensemble variability after the first few years of the simulation (i.e., all ensemble members drift almost immediately). In the first 100 years of the simulation ensemble, the drift amounts to approximately 1 cm of global sea level equivalent, which is 5-10% of the median projected Greenland
contribution to sea level rise by 2100 in ISMIP6 (Goelzer et al., 2020a). At the end of the 2000-year simulation ensemble with highest variability amplitude (yellow line), the drift is larger than 1.5% of total initial ice mass, or about 12 cm of sea level equivalent. Based on these two ensembles, we conclucde that the rate of drift increases with the amplitude of the variability in calving rate. As a point of comparison, the dashed line shows a single simulation, without stochastic variability, but with a 270% increase in the mean calving rate at all 11 marine terminating glaciers for which we simulate terminus migration. The
spatial pattern of ice thickness change in this simulation (not plotted) is very similar to the stochastic calving ensemble with highest variability amplitude, indicating that the noise-induced drift in the stochastic ensemble occurs due to increased mass loss at the terminus. This indicates that model drift due to a realistic level of noise in just the annual calving rate is equivalent to ice loss from a substantial increase in calving rate without noise. Calibrating a deterministic model to match the observed ice sheet state, which is subject to variability from climatic and glaciological processes, would require tuning parameters to
very different values. We discuss the resulting biases in section 4.

      Variability in SMB (green line) does not drive discernible drift in the ice sheet volume, in contrast to the study of Lauritzen et al. (2023) which found strong noise-induced drift in an ensemble of Greenland ice sheet simulations in response to temperature variability applied through a positive-degree-day model. We do not use such a model to parameterize SMB. Instead, we specify stochastic variability directly in SMB on a catchment-by-catchment basis.

While these stochastic ensembles exhibit less than 2% changes in their total Greenland ice sheet mass after 2000 years, the local change in ice thickness at some of the largest marine-terminating glaciers in Greenland is a substantial fraction of their initial ice thickness (Figure 2c). At some glaciers, there is thinning in some ensemble members and thickening at others. At

other glaciers, all ensemble members show thinning. To show the expression of this noise-induced drift at different glaciers, we further plot profiles of ice thickness for all ensemble members at Sermeq Kujalleq (also called Jakobshavn Isbrae) in

Figure 2a-b and Petermann Glacier in Figure 2d-e. Under a sufficiently large amplitude of variability in calving rate, retreat of the terminus of Sermeq Kujalleq occurs episodically with timing that is variable across ensemble members (Figure 2a). At Petermann Glacier, retreat of the terminus is monotonic and nearly uniform across ensemble members during the early parts of simulations (Figure 2d-e). The different expressions of this drift indicate that there is likely to be more than one mechanism responsible for producing the drift, as explored in the next section.

## 130   3   Causes of Noise-Induced Drift in Ice Sheets

Many systems, including the climate system (Penland, 2003), exhibit noise-induced drift, wherein inclusion of noise causes a change in the mean system state. To explain the potential causes of noise induced drift, we start from a generic stochastic differential equation

$$\frac{dx}{dt} = f(x) + g(x)\eta(t)^{\beta}, \tag{3}$$

where $x$ is the model state, $t$ is time, $f(x)$ is a function describing the deterministic model dynamics, $g(x)$ is a function describing how the amplitude of the noise forcing the system may depend on model state, $\eta(t)$ is a random noise term drawn from some distribution (typically Gaussian), and $\beta$ is an exponent. For the sake of simplicity, we treat Eq. (3) in a scalar form, but it can be generalized to a vector-valued case without loss of generality. In the case where $f(x) = -\alpha x$, $g(x) = 1$, $\beta = 1$, and $\eta(t)$ is a random variable drawn from a Gaussian distribution, this is the Langevin equation describing Brownian motion

of a particle without drift. However, in many more complex systems, real physical processes described by the components of this equation lead to noise-induced drift. For a more technical review of noise-induced drift, the interested reader is referred to Horsthemke and Lefever (1984).

Here, we describe three causes of noise-induced drift that are potentially relevant to ice sheets:

1. **Noise-induced bifurcation/tipping:** In Equation 3, when $f(x) = \alpha x$, $\alpha$ describes the linear stability of the system. If $\alpha$

is negative the system is stable, as perturbations from the noise term $\eta(t)$ will be damped. If $\alpha$ is positive the system is unstable as perturbations from the noise term $\eta(t)$ will not be damped. Thus, if a noise perturbation causes $\alpha$ to change sign (i.e., a bifurcation), the system will undergo a transition to a different state. Such stability properties have been previously explored in the context of ice sheet dynamics where loss of ice sheet stability through the "marine ice sheet instability" or other bifurcations, may be caused by variability in climate forcing (Mulder et al., 2018; Christian et al.,

2022; Sergienko and Haseloff, 2023).

2. **Multiplicative noise:** In Equation 3, when $g(x)$ is any function that is not even about the fixed point $x_*$ ($\frac{\partial f}{\partial x}|_{x=x_*} = 0$), i.e., $g(x_* - \eta) \neq g(x_* + \eta)$. This describes any system where the amplitude of noise perturbations depends on the system state, causing the entire noise term $g(x)\eta(t)$ to have a non-zero mean. Physically, such multiplicative noise arises in

systems where there is noise in a term that depends on system state. This has previously been explored in the context of simple glacier models (Mantelli et al., 2016; Robel et al., 2018; Mikkelsen et al., 2018).

3. **Nonlinear or asymmetric noise:** If $\beta \neq 1$ (excluding the trivial case where $\beta = 0$) or if the underlying noise process has non-zero mean (i.e., the distribution of noise is intrinsically "asymmetric"), then the noise term will cause drift in the mean system state. Because most canonical stochastic models assume that the noise term is linear and sampled from a Gaussian distribution, this potential cause of noise-induced drift has received considerably less attention in the literature (although discussed in detail by Horsthemke and Lefever, 1984). Glacier ice is a viscous non-Newtonian fluid, meaning that glacier flow speed exhibits a strong nonlinear sensitivity to many different types of forcing (Glen, 1955; Millstein et al., 2022). Robel et al. (2018) previously considered this source of noise-induced drift in the context of ice shelf buttressing, but many other processes related to ice flow may exhibit similar nonlinear-noise-induced-drift.

To understand the role of these different potential causes of noise-induced drift in ice sheet dynamics, we consider several highly idealized stochastic ensembles. In each simulation, we use StISSM to simulate ice velocity and thickness evolution of a single marine-terminating glacier in a rectangular channel of uniform width, without floating ice. Model configuration choices such as the stress balance approximation and the basal sliding law are identical to the Greenland ensemble described in the previous section, but with spatially uniform basal friction coefficient ($C^2$). In all configurations, an initial deterministic steady-state is obtained by holding all forcing variables constant and running until the total ice mass of the glacier changes by less than 0.05% in 200 years. In each idealized stochastic ensemble, calving rate is drawn from a Gaussian distribution (i.e., white noise) with mean equivalent to the initial deterministic calving rate and standard deviation equal to 1/3 of the mean. We perform ensemble simulations of 30 members each, running for 2000 years.

### 3.1 Noise-induced bifurcation/tipping

Figure 3 shows the results of three idealized stochastic ensembles, all of which have the same background prograde slope of 0.004 in bed topography. In the first stochastic ensemble (Figure 3a-b), the bed topography includes a single sinusoidal bump of 100 m height in bed topography at the initial terminus position. Once stochastic calving begins, 95% of the ensemble members start retreating past the bump within the first 140 years of the simulation. The second ensemble (Figure 3c-d) is identical to the first, except without a bump in bed topography, and the steady-state calving rate used in the spin-up is adjusted to maintain a similar terminus position. Though the initial glacier state is not identical due to the difference in bed topography, it is sufficiently similar that we do not attribute the subsequent behavior to a different glacier state. Instead of retreating, all ensembles members advance in response to stochastic calving. The different response to stochastic forcing between these two ensembles indicates that the ensemble-mean retreat in the first ensemble is caused by the presence of the bump in bed topography, which adds a well-understood bifurcation to the system dynamics related to a positive feedback in ice flow with bed depth. This provides a simple example of mechanism #1 identified above: noise-induced bifurcation/tipping.

When a noise-induced bifurcation drives drift in the mean-state, the rate of drift will depend on the amplitude of stochastic variability up to the amplitude of variability necessary to drive all ensemble members across the bifurcation with high prob-

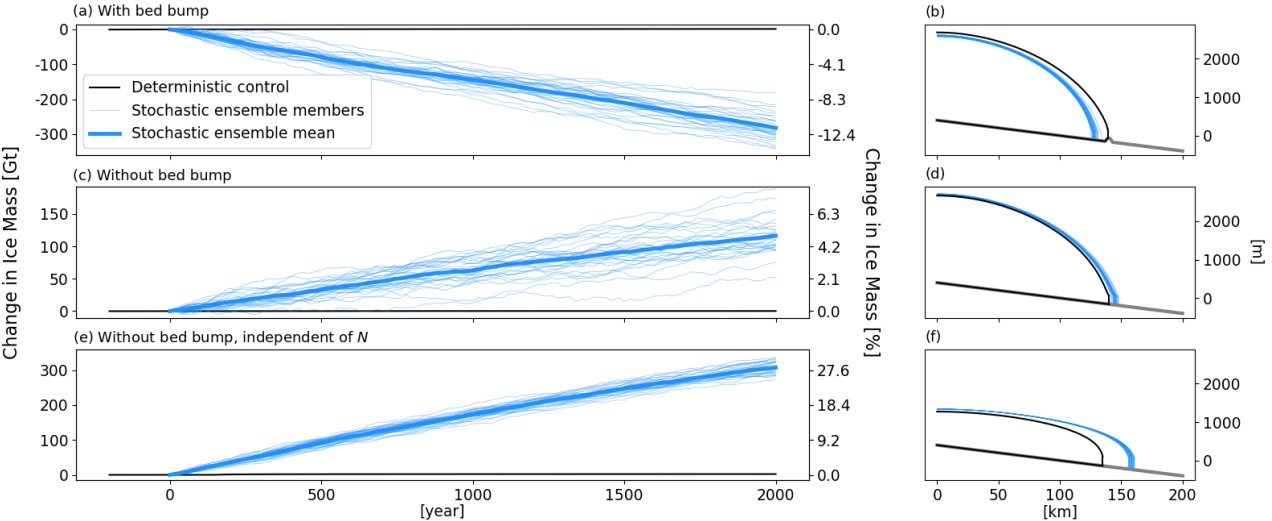

**Figure 3.** Stochastic ensembles for an idealized marine-terminating glacier in a rectangular channel on a prograde bed slope. (a-b) Including a sinusoidal bed bump. (c-d) Without bed bump. (e-f) Without bed bump and the effective pressure constant in time. Left panels show change in ice mass over time, right panels show glacier thickness profiles at the end of simulations. In all panels, black line is deterministic control run, thick blue line is stochastic ensemble mean, thin blue lines are all stochastic ensemble members.

ability. In Figure 3a-b this "saturation" of drift rate is occurring as all ensemble members eventually cross the bifurcation. Further increasing the amplitude of the variability will not be able to drive more ensemble members through the bifurcation, though they might reach it faster near the beginning of the simulation, causing faster initial drift of the mean-state. As a point of comparison, in the full Greenland ice sheet ensemble discussed in section 2, the magnitude of ensemble-mean retreat at Sermeq Kujalleq shows clear dependence between low (Figure 2b) and high (Figure 2a) amplitude of calving variability. In this case, further increasing the amplitude of variability may cause some ensemble members to retreat past the second bed peak, thus increasing the extent of noise-induced drift.

### 3.2 Multiplicative noise

Noise-induced tipping is clearly not the only mechanism causing the drift seen in the more realistic simulations discussed in the prior section, since drift still occurs even in the absence of a bifurcation in system dynamics. Multiplicative noise (mechanism #2) may explain this drift in the second stochastic ensemble as variability at the calving front perturbs the near-terminus thickness, causing variations in effective pressure and ultimately velocity through the Budd sliding law (Eqs. (1)-(2)). This particular sliding law includes a linear dependence of basal friction on effective pressure, and therefore ice thickness, though there are other nonlinearities elsewhere which may play a role in generating drift. Since the variable that is being perturbed stochastically is linearly related to ice flow, and the nonlinearities arise elsewhere in the ice sheet dynamical equations, this is considered to be "multiplicative noise", similar to $g(x)$ being multiplied by $\eta(t)$ in Eq. (3). To investigate this possibility, we

consider a stochastic ensemble (Figure 3e-f) in which the effective pressure dependence is removed from Eq. (1), effectively introducing a sliding law linear in sliding velocity only. In this case, drift still occurs, indicating that multiplicative noise through evolving effective pressure is unlikely to be the only mechanism causing the drift. Though ice sheet dynamics involve the complex interplay of many factors, the lack of other obvious multiplicative feedbacks likely to cause a significant asymmetry in the variability of terminus thickness or velocity strongly indicates the drift seen in these two ensembles is mainly caused by a different mechanism: nonlinear noise (mechanism #3 above).

### 3.3 Nonlinear noise

Though there are many sources of nonlinearity in ice sheet dynamics, the fact that only stochasticity in calving causes drift in the Greenland ensemble of the previous section indicates that it is some nonlinear process specific to the glacier terminus which leads to noise-induced drift in the absence of a bifurcation. Here we give mathematical explanations for the drift in response to stochastic variability in terminus position applying to tidewater glaciers and glaciers with floating ice shelves.

In all stochastic simulations considered in this study, the mean of the rate of calving at the terminus ($u_c$) does not change, and so any changes in the time-averaged terminus position must be the result of changes in mean ice flow velocity towards the terminus ($u_f$). For a tidewater glacier, like that simulated in Figure 3, $u_f$ is determined by the momentum balance at the terminus

$$2hA^{-1/n}\left|\frac{\partial u_f}{\partial x}\right|^{1/n-1}\frac{\partial u_f}{\partial x} = \frac{\rho_i g}{2}\left(h^2 - \lambda b^2\right) \tag{4}$$

where $h$ is the terminus thickness, $b$ is the water depth, $\rho_i$ is the ice density, $\lambda = \frac{\rho_w}{\rho_i}$ is the ratio of water to ice density, $g$ is the gravitational acceleration, $A$ is the depth-integrated Glen's flow law rate factor, and $n$ is the Glen's flow law exponent. Perturbations to the mean terminus position may cause perturbations to the glacier thickness and bed depth at the terminus which can be included through a Reynolds decomposition: $h = \langle h \rangle + h'$ and $b = \langle b \rangle + b'$, where all variables enclosed with $\langle \rangle$ are time-averaged and perturbed variables are denoted by $'$. All perturbed variables are drawn from a Gaussian distribution with zero mean. Including these decomposed expressions into the above momentum balance and simplifying yield an expression for the strain rate at the terminus in terms of perturbations

$$\frac{\partial u_f}{\partial x} = \frac{A\rho_i^n g^n}{2^{n+1}}\left(\langle h \rangle + h' - \lambda\frac{(\langle b \rangle + b')^2}{\langle h \rangle + h'}\right)^n. \tag{5}$$

The quadratic term in this expression is expanded, and we separate terms with only the mean state in their numerator from those including perturbations in their numerator

$$\frac{\partial u_f}{\partial x} = \frac{A\rho_i^n g^n}{2^{n+1}}\left[\left(\langle h \rangle - \frac{\lambda\langle b \rangle^2}{\langle h \rangle + h'}\right) + \left(h' - \frac{2\lambda\langle b \rangle b'}{\langle h \rangle + h'} - \frac{\lambda b'^2}{\langle h \rangle + h'}\right)\right]^n. \tag{6}$$

We perform a Taylor expansion on the resulting expression in terms of the exponent $n$ keeping in mind that terms involving perturbations will generally be smaller than terms involving only the mean state. Thus, terms depending on higher powers of $h'$ and $b'$ can be neglected, and we only keep the first two terms of the expansion (i.e., linearizing):

$$\frac{\partial u_f}{\partial x} = \frac{A\rho_i^n g^n}{2^{n+1}}\left[\left(\langle h \rangle - \frac{\lambda\langle b \rangle^2}{\langle h \rangle + h'}\right)^n + n\left(\langle h \rangle - \frac{\lambda\langle b \rangle^2}{\langle h \rangle + h'}\right)^{n-1}\left(h' - \frac{2\lambda\langle b \rangle b'}{\langle h \rangle + h'} - \frac{\lambda b'^2}{\langle h \rangle + h'}\right)\right]. \tag{7}$$

We re-arrange this expression to emphasize the relative influences of the mean state and perturbations

$$\frac{\partial u_f}{\partial x} = \frac{A\rho_i^n g^n}{2^{n+1}} \left( \langle h \rangle - \frac{\lambda \langle b \rangle^2}{\langle h \rangle + h'} \right)^n \left[ 1 + n \left( \langle h \rangle - \frac{\lambda \langle b \rangle^2}{\langle h \rangle + h'} \right)^{-1} \left( h' - \frac{2\lambda \langle b \rangle b'}{\langle h \rangle + h'} - \frac{\lambda b'^2}{\langle h \rangle + h'} \right) \right]. \tag{8}$$

To understand the effect of perturbations on the glacier mean state we take a time average of this expression, which eliminates terms that are linear in a perturbation variable because they have a mean of zero

$$\left\langle \frac{\partial u_f}{\partial x} \right\rangle = \frac{A\rho_i^n g^n}{2^{n+1}} \left( \langle h \rangle - \frac{\lambda \langle b \rangle^2}{\langle h \rangle} \right)^n \left[ 1 - \frac{n\lambda \langle b'^2 \rangle}{\langle h \rangle^2 - \lambda \langle b \rangle^2} \right]. \tag{9}$$

Note in the above step, terms which include perturbations as a sum in the denominator, are linearized through a Taylor series expansion, before the average is taken, leaving only the terms involving the mean state, $\langle h \rangle$. If the perturbation terms are drawn from a Gaussian distribution with variance $\sigma^2$, then terms involving the square of the perturbation are drawn from a gamma distribution $\Gamma\left(\frac{1}{2}, 2\sigma^2\right)$, which has a non-zero mean equal to $\sigma^2$. Thus, the rate of drift depends on how large $n\lambda\sigma_b^2$ is relative to $\langle h \rangle^2 - \lambda \langle b \rangle^2$, where $\sigma_b^2$ is the variance of the perturbations in bed depth due to perturbations in ice front position. The sign of this leading order term causing the drift is negative, causing a decrease in the near-terminus strain rate, and a net positive mass balance near the terminus, driving advance. While we might expect that $\sigma_b << \langle b \rangle$, if the bed topography ($b_x$) is steep, then $\sigma_b = b_x \sigma_L$ (where $\sigma_L$ is the standard deviation of variability in terminus position) could be a non-negligible fraction of $\langle b \rangle$, causing appreciable drift. Also, if the terminus is at or near flotation, then $\langle h \rangle^2 - \lambda \langle b \rangle^2 \approx \lambda^2 \langle b \rangle^2 - \lambda \langle b \rangle^2 \approx 0.1 \langle b \rangle^2$ and the denominator of the above expression would be sufficiently small to admit non-negligible drift. The simulations in Figure 3c-f do exhibit such thickening and advance of the initially grounded terminus. Given that both steep bed topography and near-flotation termini are common in Greenland, we may expect this effect to be common, though we do not simulate any cases of ensemble-mean glacier advance in the more realistic Greenland ensemble (Figure 3c).

For a glacier with a floating ice shelf, the calving front is not grounded and so the momentum balance does not depend on the bed depth, making the above analysis not applicable. Rather, we consider the effect of buttressing from the floating ice shelf on the velocity of ice through the grounding line. Haseloff and Sergienko (2018) perform an asymptotic analysis to derive an approximation for the ice flow velocity ($u_g$) through a strongly buttressed grounding line

$$u_g = \left[ \frac{(1 - \lambda^{-1})\rho_i g}{(1 + n^{-1})\Lambda L_s} \right]^n h_g^n \tag{10}$$

where $\Lambda$ is a parameter governing lateral shear stress within the ice and $L_s$ is the ice shelf length. This expression assumes that ice loss occurs entirely through ablation at the calving front and lateral shear stresses increase linearly across the ice shelf. We consider stochastic calving at the calving front of the floating ice causing Gaussian, zero-mean perturbations to the ice shelf length: $L_s = \langle L_s \rangle + L_s'$. We insert this Reynolds decomposition in the above expression for grounding line flux, and take the Taylor expansion of the resulting expression

$$u_g = \left[ \frac{(1 - \lambda^{-1})\rho_i g}{(1 + n^{-1})\Lambda} \right]^n h_g^n \left( \langle L_s \rangle^{-n} - n\langle L_s \rangle^{-n-1} L_s' + \frac{n(n+1)}{2} \langle L_s \rangle^{-n-2} L_s'^2 + \dots \right). \tag{11}$$

From this expression, we neglect higher-order terms and rearrange to resemble the original flux expression

$$u_g = \left[ \frac{(1 - \lambda^{-1})\rho_i g}{(1 + n^{-1})\Lambda \langle L_s \rangle} \right]^n h_g^n \left( 1 - n\langle L_s \rangle^{-1} L_s' + \frac{n(n+1)}{2} \frac{L_s'^2}{\langle L_s \rangle^2} \right). \tag{12}$$

Taking the time-average, the term which is linear in $L_s'$ vanishes, leaving

$$\langle u_g \rangle = \left[ \frac{(1 - \lambda^{-1})\rho_i g}{(1 + n^{-1})\Lambda\langle L_s \rangle} \right]^n h_g^n \left( 1 + \frac{n(n+1)}{2} \frac{\langle L_s'^2 \rangle}{\langle L_s \rangle^2} \right). \tag{13}$$

The $L_s'^2$ term is drawn from the $\Gamma\left(\frac{1}{2}, 2\sigma_{L_s}^2\right)$ distribution, which has a non-zero mean equal to $\sigma_{L_s}^2$. Thus, when $\frac{n(n+1)\sigma_{L_s}^2}{2}$ is non-negligible compared to $\langle L_s \rangle^2$, the time-averaged ice flow velocity through the grounding line is increased by stochastic calving, which will cause the grounding line to retreat. We can note here that different assumptions can be made about the form

of lateral shear stress variation across the floating ice shelf or the dominant source of mass loss, and in general the rate of ice flow through the grounding line will be nonlinear in terms of the ice shelf length (Haseloff and Sergienko, 2018), causing the sort of nonlinear noise-induced drift discussed here.

### 3.4    Attributing causes of drift

Returning to the more realistic Greenland ice sheet ensemble simulations (Figure 2a-b,d-e), we conclude that in most glaciers
for which strong noise-induced drift is simulated, there are easily identifiable bed topography features indicating that noise-induced bifurcations are the most common cause of noise-induced drift (as previously argued in Christian et al., 2022). Conversely, there are no tidewater glaciers in this realistic ensemble exhibiting ensemble-average terminus advance due to nonlinearity in hydrostatic stress terms discussed in the previous section. This is likely because glaciers tend to stabilize at peaks in bed topography (Robel et al., 2022), making it more likely that the sudden onset of stochastic calving would lead to a retreat
from noise-induced bifurcation, rather than sustained advance due to the nonlinear-noise mechanism. In contrast, during the earliest stage of Petermann Glacier's retreat (Figure 2e), the bed is entirely prograde and yet ensemble-mean retreat still occurs. At the time of our study, Petermann Glacier is one of only two glaciers in Greenland with a buttressing ice shelf remaining. Thus, the mechanism of drift due to nonlinearities in buttressing, discussed in the previous section, is likely responsible for the early stages of strong retreat of the Petermann grounding line, before reaching a bed peak, after which a noise-induced
bifurcation over a bed peak is likely also playing an important role in the simulated ensemble-mean retreat.

     We also note briefly that Lauritzen et al. (2023) find that variability in surface temperature can cause noise-induced drift through a positive-degree-day (PDD) model for SMB, though they do not speculate on the cause of this drift (or refer to it as such). It is likely that the strong nonlinearities in their PDD model are the cause of the noise-induced drift they find in their results, as their simulations do not appear to include bifurcations in SMB or sources of multiplicative noise. Regardless of
the precise mechanism of noise-induced drift in different model configurations, our simulations show that there are a range of different mechanisms intrinsic to ice sheet dynamics that cause noise-induced drift to be an expected and essential aspect of ice sheet evolution, and therefore of realistic model simulations. We purposely adopt a conservative approach to applying stochastic forcing directly to terms in the mass conservation equations of the ice sheet model, but we expect that stochastic variability in climatic and glaciological processes drives noise-induced drift through many different mechanisms in more realistic ice sheet
simulations.

## 4 Implications for Ice Sheet Modeling

The Greenland ensemble simulations in this study exhibit a tendency for noise-induced retreat and ice loss. Thus, spin-up of an ice sheet model without variability in forcing is likely to lead to a modeled ice sheet that is biased compared to observations of real ice sheets, which are naturally subject to variable forcing and resulting noise-induced drift. Such a mismatch is typically reduced by tuning, or optimizing model parameter values, including those related to ice sliding, viscosity, calving and ocean melt, through inversion. However, calibrating a parameter to minimize model-observation mismatch arising due to processes not represented in the model may introduce compensating errors in the modeled state. Ice sheet models that tune one parameter to reduce biases in other parameters have been shown to have substantially biased sensitivity to changes in forcing (Berends et al., 2023).

Many contemporary projections of future ice sheet evolution omit variability in forcing for transient projections due to challenges related to modeling ocean circulation near ice sheets or the lack of output from climate models far into the future (Golledge et al., 2015; DeConto et al., 2021). Such an omission may lead the modeled ice sheet sensitivity to future changes to be biased, as noise-induced retreat is an important and realistic component of the forced response. As discussed in the prior section and prior studies (Christian et al., 2022), in the absence of variability, many glaciers may not cross important thresholds to rapid retreat and thus their projected response to climate forcing would be considerably less than is likely in reality. Additionally, potential future changes in the amplitude of variability (e.g., Bintanja et al., 2020) could increase the likelihood of crossing noise-induced bifurcations, and amplify the impacts of state-dependent and nonlinear noise. Such effects cannot be captured if variability in forcing is omitted entirely.

Other contemporary projections of future ice sheet evolution (e.g., many of the models participating in the recent ISMIP6 intercomparisons; Goelzer et al., 2020b; Seroussi et al., 2020) start from a calibrated initial state, and then simulate the free-running ice sheet state in response to forcing including variability. In such a simulation design, the sudden onset of variability could introduce a transient noise-induced drift. If the drift causes ice loss, as in the ensembles described in section 2, this would cause the projected ice sheet sensitivity to forcing to be too high. Other recent modeling studies use a calibrated initial state, but then re-calibrate the ice sheet sensitivity to a changing mean climate with historical observations of ice sheet change (e.g., Nias et al., 2019; DeConto et al., 2021). In such a case, the calibrated sensitivity to changes in the mean climate would be too low, due to the spurious influence of noise-induced drift following the sudden onset of variability in the model. Similarly, the practice of removing "control" simulations with forcing held constant to diagnose ice sheet sensitivity to forcing (Seroussi et al., 2020; Goelzer et al., 2020b), may introduce bias due to the lack of noise-induced drift in control simulations.

Noise-induced drift in ice sheets should not only be thought of as a source of bias in models. Real ice sheets are subject to stochastic variability in many processes, thus meaning that their state, whether steady or not, includes the effect of noise-induced drift. The potential ice sheet model biases identified here all result from an incomplete representation of these real sources of variability within climate or glaciological processes. To eliminate or lessen these biases in ice sheet models, we recommend two possible solutions for initializing ice sheets model simulations: (1) initializing directly from observed ice sheet state without relaxation, even when the ice sheet is out-of-balance, or (2) including internal variability in the forcing of

ice sheet models during spin-up. The first proposed solution recognizes that the observed state of ice sheets in the real-world subject to variability should implicitly include the tendency resulting from noise induced drift. Ice sheet modelers may prefer using such a solution as it requires less computational resources, however data assimilation methods for accurately reproducing observed non-steady ice sheet states are still a nascent area of development (Goldberg and Heimbach, 2013; Choi et al., 2023). The second suggested solution is likely to be necessary if an initial steady-state for a simulation is desired and observations of ice sheet state and tendency are not available, as in most simulations starting prior to the satellite era. Improving both glaciological process models (e.g., hydrology and calving) and the efficiency of coupling to climate models, should also yield improvements in the complete and accurate representation of variability. Finally, stochastic ice sheet modeling (e.g., StISSM; Verjans et al., 2022) provides a parallel approach to accurately include variability within ice sheet models in a computationally efficient manner.

*Code and data availability.* StISSM is an open-source large-scale stochastic ice sheet model that is currently included as part of the public release of ISSM. The public SVN repository for the ISSM code can be found at https://issm.ess.uci.edu/svn/issm/issm/trunk and downloaded using username "anon" and password "anon". The documentation of the code version used here is available at https://issm.jpl.nasa.gov/documentation/ (last access: 30 October 2023). Scripts for the Greenland spin-up simulation follow Verjans et al. (2022) and are available at: https://doi.org/10.5281/zenodo.7144993

*Author contributions.* AAR conceived the study, developed the theory, and led writing of the manuscript. VV performed the model simulations, post-processed the model output, and contributed to writing. AAA contributed to the mathematical analysis and writing.

*Competing interests.* The contact author has declared that none of the authors has any competing interests.

*Acknowledgements.* This study and all authors are a part of the Stochastic Ice Sheet Project, funded by the Heising-Simons Foundation. Thanks to John Erich Christian and Judith Berner for fruitful discussions during the conception of this study. Thanks to Johanna Beckmann for pointing out an error in the original pre-print.

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
