# Peer review of "Biases in ice sheet models from missing noise-induced drift"

_EGUsphere, 2023_

## Author Comment (AC1)

To the reviewer, thank you for your thoughtful comments. We have provided an initial comment-by-comment response below. A completed revision is nearly complete (less some in progress simulations, detailed below) and will be submitted upon solicitation by the editor. Below please find original review comments in blue and our responses in black.

**Reviewer 1**
The importance of noise and its ability to change the mean state of an ice sheet model is a relatively novel research direction and this manuscript makes important contributions to further our understanding of its role. The methods used in this manuscript are sound, but I think the conclusions in this manuscript would be strengthened with more modeling results. As it is, I do not agree that all conclusions are supported by results (see more detailed comments below). I also think that the paper would benefit from some restructuring and editing, as it appears a bit disjointed at the moment. In particular, I would move sections 3-3.3 before section 2, rather than after, and extend the abstract to be more informative.

Thank you to this reviewer for their helpful suggestions. As the reviewer can see we have made revisions, are in the process of adding more simulations as suggested, and made responses to their suggestions. The abstract has been expanded and we have been more careful not to over-generalize from our results. We have started some of the suggested new simulations (the ice sheet wide ensembles take about 3 weeks to run) which do not change the results in a substantive way. In other places we were able to mitigate the need for new simulations by clarifying the text.

As for the suggestion to move sections 3-3.3 in front of section 2, while this may make more sense from the point of view of establishing theory first and then showing a more realistic large-scale simulation result, part of the goal of this study is to get readers to buy-in that noise-induced drift is an issue for realistic ice sheet projections, which is why we put the large-scale ensembles first before explaining the theory behind the results. Also, the way the discussion has been structured, we refer back to the large-scale simulations as examples of these different mechanisms of noise induced drift in section 3, and this would not be possible if the theory/idealized model sections come first. We strongly feel that the order of present a result that needs explaining-and then explaining it after is an equally valid way to structure the paper, but if the reviewer has an idea of how to restructure in a way that preserves the current mode of discussing (without needing to cut significant portion of the discussion), then we would be open to it.

However, we think you will find that the edits that have been made greatly improve the clarity in the ways suggested and the revised manuscript is improved as a result.

**Abstract:** To me, the abstract seemed too short and not very informative. It would be more informative if it included more details about the methods used. Also, it would be good to point out that stochastic variability in frontal ablation changes the mean state more than variability in other parameters.
We have added further details in the abstract, specifically describing the methods used, the results for stochastic variability in SMB, and the different causes of noise induced drift.

**Introduction:**

The introduction is quite a bit more general than the abstract which refers to frontal ablation. It's almost as if the purpose of the paper changed from abstract to introduction. Please streamline. This perception of the reviewer is probably the result of this paper originally being written as a brief communication, and then expanded. However, our expansion of the abstract, and then our inclusion of more specific wording in the introduction make the two parts feel more in tune with each other.

Line 21: "we show that noise-induced drift is expected in [...] any numerical modeling of ice sheets." I am pretty sure it is possible to construct counter-examples which do not show noise-induced drift, so you might want to be careful with such sweeping statements.
Changed to be less absolute: "noise-induced drift is expected to occur in real ice sheets and numerical modeling of ice sheets"

**Section 2:**
The variability in water pressure is quite a bit smaller than the variability in calving and only specified in an unspecified "region near the glacier front", which makes a direct comparison with calving and SMB variability difficult. I recommend either including more information or leaving it out completely.
We have removed this ensemble completely, as it was challenging to make a direct comparison with the other ensembles, as this reviewer points out.

Line 75: "the rate of drift is also approximately proportional to the amplitude of the variability in calving rate": Fig. 1 does not show this. The figure shows that the drift observed with stochastic calving appears to increase with the noise variability (interpolating from 2 data points) and that the mass change induced by stochastic noise might be reproduced by a higher deterministic calving rate. More data examples would be required to make more definite statements. If feasible, I highly recommend including more data in order to analyse the behaviour in more detail.
This is a good point, and one that perhaps we were a bit too expansive in claiming. We have softened the language here and then we have added a discussion of the question of drift proportionality to stochastic forcing amplitude in section 3. Adding further ensembles would be computationally expensive, and not necessary to accomplish the principle goals of this paper.

It would also be interesting to see what the model behaviour for stochastic noise in calving, SMB and pw combined is. Are drifts due to different processes simply additive, or might there be some nonlinear bahaviour?
This is a good suggestion. We had done some initial short simulations which showed that the processes are additive (though drift is dominated by calving forcing). We have decided to re-run these completely, and the combined ensemble will be included in the revised manuscript.

Figure 1 legend: what does St. stand for? I think writing "stochastic" out would be clearer.
Changed

**Section 3:**
Equation: personally, I would prefer f and g to be bold too, to be consistent with x as a vector
Changed

Line 158-160: The effective pressure used in this study should be N=ρi g H + ρw g b. It would be useful to write this out here, so that readers not familiar with this particular choice of sliding law are immediately aware of the dependence of N on H. While I see how N varies linearly with changes in ice thickness, it does not necessarily follow that variations of the terminus position lead to linear variations in ice thickness or that there is a linear dependence of u on H; please rewrite this to be clearer.'

We added the definition of effective pressure back in section 2 where the model is explained, and then referred back to this equation in section 3. We also removed "linear" to avoid confusion about the point here, which is simply that velocity at the front is indirectly effected by terminus fluctuations through the effective pressure and sliding law.

More generally in section 3.2 it is not clear why the non-linear dependence of u on H leading to drift is fundamentally different from the case considered in 3.3. To me, the only difference is that the effect on effective pressure is located away from the terminus, which makes it less amenable to a Reynolds decomposition - and which would also explain the potentially smaller effect of these perturbations.

The dependence of basal friction on effective pressure is linear in the Budd sliding law considered here. It is true that there are other nonlinearities in the momentum balance that could lead to drift, but the distinction that is made in the statistical physics literature is whether the asymmetry arises directly in the variable that is being stochastically perturbed (termed "nonlinear noise") or due to state-dependence (whether linear or not) on other system variables ("multiplicative noise"). While this may seem a bit of a semantic distinction, particular in systems as complex and nonlinear as ice sheets, our goal here is to introduce the glaciology community to the way that this problem is thought about in the physics and math literature, and so we see it as important to preserve this distinction. We have added further clarification to make the distinction clearer:

"In this particular sliding law, the dependence of basal friction on effective pressure (and therefore ice thickness) is linear, though there are other nonlinearities elsewhere which may play a role in generating drift. Since the variable that is being perturbed stochastically is linear, and the nonlinearities arise elsewhere in the ice sheet sheet dynamical equations, this is considered to be 'multiplicative noise'."

Line 211: though -> through
Fixed

The analysis in section 3.3 makes specific predictions about the size of noise-induced drift depending on σ, and . It would be helpful to verify and illustrate these results with the idealized marine-terminating glacier model.

We are currently running new idealized simulation ensembles with changes in the amplitude of variability to show this point. We anticipate including these simulations in the revised manuscript.

---

## Author Comment (AC2)

To the reviewer, thank you for your thoughtful comments. We have provided an initial comment-by-comment response below. A completed revision is nearly complete (less some in progress simulations, detailed below) and will be submitted upon solicitation by the editor. Below please find original review comments in blue and our responses in black.

**Reviewer 2**

The manuscript describes ice sheet model experiments and a theoretical analysis that deal with the issue of model drift introduced by including stochastic forcing in simulations. The analysis shows that noise-induced model drift is to be expected for model setups that were initialised with deterministic forcing and then activate stochastic forcing in forward experiments. It also suggests that other setups could be impacted by the absence of realistic stochastic forcing. The paper is well written and presents a thorough analysis of the problem, albeit without offering an approach how to make meaningful projections under stochastic forcing.

Thank you to the reviewer for offering such thoughtful critiques of the paper. As they will find, we have addressed their main concern by re-writing the ending of the paper to be much clearer about the implications of this work and two concrete solutions for addressing the issues raised here. A long response to that particular point can be found below general point 2 below.

General comments

1. In the context of this study, it seems important to discuss what a 'realistic stochastic forcing' (l23, l50) actually is (in amplitude and temporal variability) and at what level that forcing is symmetric. It appears that the existence of a drift and its amplitude could crucially depend on how exactly the forcing is parameterised. In case of SMB, there is the suggestion that symmetry in temperature variability could translate through a non-linear SMB model (e.g. PDD) to an SMB forcing with an asymmetric variability. This would mean that results depend on the level at which symmetry in the forcing is prescribed. Could the same be true if instead of perturbing ocean thermal forcing or calving, a higher level ocean forcing, like advance/retreat is randomised symmetrically? While this is interesting for the interpretation of case 2 (Multiplicative noise) and case 3 (Nonlinear or asymmetric noise) in the theoretical analysis, I would think it is especially important for finding a practical approach to initialising a stochastic ice sheet model.

As discussed below, we have further expanded the discussion of prior studies to argue that standard deviation being 1/3 of the mean in SMB and frontal ablation forcing is realistic. The point about the "level" at which stochasticity is applied is a good one. In a sense, the approach we have taken here is the simplest one, and one where the symmetric stochasticity is applied at the lowest level, directly to terms which appear in ice sheet evolution equations: SMB being a source term in the mass conservation equation and frontal ablation being a sink term in the terminus level set equation (in ISSM). Constructing our experiments in this way means that we can be sure that when drift occurs it is due to ice sheet dynamics, rather than choice of SMB or ocean melt parameterizations. However, this is an excellent point to make, and one that we have added further discussion of at this point in the manuscript. Below is the new paragraph:

"In implementing white noise forcing in SMB and frontal ablation rate, we introduce symmetric variability directly in terms of the mass conservation equations for the ice sheet. This simplifies the task of identifying potential causes of resulting noise-induced drift, since the only dynamics to consider are those related to ice sheet flow. However, it may be that in reality, symmetric variability occurs in variables more removed from ice sheet dynamics such as atmospheric or

ocean temperatures. Then, asymmetries or nonlinearities in the dependence of mass fluxes on these variables can be an additional source of noise-induced drift, as previously discussed by Mikkelsen et al. (2018) and Lauritzen et al. (2023). Our goal in this study is to identify mechanisms of noise-induced drift that are inherent in the fundamental dynamics of ice sheet flow. Such mechanisms would likely be common to all ice sheet models, and not dependent on the model-specific parameterizations of mass fluxes as a function of climate forcing.”

2. Model drift in the mean of the simulations is shown to arise by switching from deterministic forcing in the spin-up to stochastic forcing in the forward experiments. While this is an instructive example to understand something about the effect of stochastic forcing in ice sheet simulations, it clearly shows that this stochastic model setup cannot be operated for projections. In other words, what is really missing here is an approach/recommendations/proof of concept how to initialise a stochastic model in a meaningful way. How would an initialisation look like for i) an assumed steady state some time in the past, ii) a situation of mass gain, iii) the present day state (e.g. approximately matching the ongoing mass change)? Without offering a solution, I think the paper should step away from making recommendations about the use of stochastic forcing in ice sheet simulations.

This was a helpful comment because it caused us to rewrite the final paragraph of the paper to be clearer about a few points and then to state plainly what our recommended solutions are. First off, we should be clear that the noise-induced drift we identify in this study is not spurious or considered to be a purely “unphysical” model artifact. That is, we have pointed out that an ice sheet (real or modeled) subjected to stochastic forcing will experience transient drift or added drift tendency that still exists even once a new statistical steady-state has been reached by the model. This means that observations of ice sheet state implicitly include this drift tendency (even if they are at steady-state, in this case, they’ve just moved to a different mean state which modifies the deterministic terms of their dynamics to balance the drift tendency). Thus, the potential for bias in models occurs (as the reviewer rightly identifies) only when variability is not well represented, either in the spinup or the transient simulation/projection of interest.

It is unclear to us why the reviewer concludes that “this stochastic model setup cannot be operated for projections”. Indeed, the author team has a series of papers, cited here, on just this topic (Verjans et al. 2022, 2023, Ultee et al. 2024). We can sympathize with the potential concern of modelers that it is impractical to run stochastic ice sheet models. We thought the same thing when we first happened upon this issue of noise-induced drift. To allay the potential concerns of modelers, we have also pointed out that initializing a model from observations without relaxation towards a steady-state (at constant forcing) should, in principle, also serve as a solution, since as we argue above, real ice sheets include the tendency due to noise-induced drift. The newly revised paragraph should address the concerns of the reviewer, as can be seen here:

“Noise-induced drift in ice sheets should not only be thought of as a source of bias in models. Real ice sheets are subject to stochastic variability in many processes, thus meaning that their state (whether steady or not) includes the effect of noise-induced drift. The potential ice sheet model biases identified here all result from an incomplete representation of these real sources of variability within climate or glaciological processes. To eliminate or lessen these biases in ice sheet models, we recommend two possible solutions for initializing ice sheets model simulations: (1) initializing directly from observed ice sheet state without relaxation, even when the ice sheet

is out-of-balance, or (2) including internal variability in the forcing of ice sheet models during spin-up. In practice, the first suggested solution requires less computational resources, but data assimilation methods for accurately reproducing observed non-steady ice sheet states are still a nascent area of development (Goldberg and Heimbach, 2013; Choi et al., 2023). The second suggested solution is likely to be necessary if an initial steady-state for a simulation is desired and observations of ice sheet state and tendency are not available, as in most simulations starting prior to the satellite era. Improving both glaciological process models (e.g., hydrology and calving) and the efficiency of coupling to climate models, should also yield improvements in the complete and accurate representation of variability. Finally, stochastic ice sheet modeling (e.g., StISSM; Verjans et al., 2022) provides a parallel approach to accurately include variability within ice sheet models in a computationally efficient manner.

Specific comments
Title "Biases in ice sheet models from missing noise-induced drift"
I find the title confusing as it suggests that missing drift in ice sheet models is a problem. I would characterise the drift that is realised in the presented experiments is an artefact of switching from deterministic forcing in the spin-up to stochastic forcing in the forward experiments. Maybe "Biases in ice sheet models from missing stochastic variability".
As we argue at length above, it is actually the case that the drift, or the tendency to drift, *should* be represented in ice sheet models, as we argue that it is an expected component of the real ice sheet system. As this argument is made more clearly and explicitly now both in the abstract and at the end of the paper, we opt to keep the title as is.

l1 Abstract: The abstract is quite short and could be improved by adding more information about important aspects of the paper. I am thinking about more detail on section 3 and the three causes of noise-induced drift.
We have added further details in the abstract, specifically describing the methods used, the results for stochastic variability in SMB, and the different causes of noise induced drift.

l13 "the mean state of glaciers and ice sheets". It is not clear to me what the mean state of an ice sheet is without knowing the time scale of interest. Is that diurnal, annual, decadal, centennial or millennial? Maybe remove "mean" to avoid that complication, or introduce the time scale of interest, probably multi-decadal?
We mean the long-term state of glaciers and ice sheets, as set by the multi-decadal mean and trend in climate forcing, since the response time scale of viscous ice flow is decades to millennia. We have added clarifying text here.

l25 "We close by arguing that all modern ice sheet models omitting variability in climate and glaciological processes produce biased estimates of the ice sheet mean state and the ice sheet response to climate change."That's a very strong statement that I think it needs some moderation. First, it seems difficult to come to a conclusion about all models by only looking at one. Second, there is at the very least the possibility that the bias in a projection is zero, or close to zero even if we know that a model could be biased in theory. Adding a "could" in  "processes could produce" would help to mitigate my concern.
Perhaps this was unnecessarily strong. We have modified the sentence to: "We close by arguing that modern ice sheet models omitting variability in climate and glaciological processes could

produce biased estimates of the ice sheet mean state and the ice sheet response to climate change."

The split is effective in the implementation of the forcings, as the SMB and ocean melting are different in each catchment. Ice flow is free to move/communicate in between catchments as in any typical 2D ice sheet model. Modified sentence to indicate this purpose.

The changes are very weak and spread evenly around the ice sheet (see below). We have added a sentence describing this pattern, but it does not seem like it adds much to include this figure in the manuscript.

[Figure]

We have expanded the discussion here specifically with reference to several model, reanalysis and observational studies to argue that setting the standard deviation to 1/3 the mean is in line with many lines of evidence for both SMB and ocean thermal forcing. 1/6 is thus conservatively low.

Figure 1 caption "deterministic but with calving rates multiplied by 2.7". Is the spatial pattern of this experiment similar to the results shown in Figure 2c? Would be interesting to discuss. Could compare results in figures similar to 2a,b.
Yes, the pattern is very similar (see below). We have added a discussion of this in the main text.

[Figure]

l58 "is set to just 2% of the mean ice overburden pressure" Could you explain a bit more? What values would you have liked to set the standard deviation to if you could freely choose? What would have motivated this number? Observations? (See point l55).
We have removed this ensemble completely, as it was challenging to make a direct comparison with the other ensembles.

l59 "greater levels of noise lead to numerical instability in the ice sheet model" How far away from instability is the model operated in the experiments that are deemed stable? What happens to the model when it is operated close to instability? Is the model sensitive to the sign of the perturbations, i.e. more stable with high or low water pressure levels? I think it needs some more work and words to convince the reader that none of the presented results could be influenced by running the model close to instability. I don't know of any rules of thumb, but running a model at 1% when it is known to be unstable for >2% is a precaution one could take.
As noted above, we have removed this ensemble. The issue mainly is that the numerical solver is challenged by rapid changes in terms directly in the momentum balance. That is, it has trouble converging when starting from the solution at the last time step. It may be that there is a better way to implement such a stochastic forcing numerically (or it could just be that this particular ice sheet model has trouble with such forcing), but this study isn't really the place for dealing with such issues.

L61 "observe the immediate ice sheet response" Not sure many people would agree 2000 years is the immediate response. Maybe "to observe which state the ice sheet evolves towards".
Good point. We have modified this to "…to observe the ice sheet evolution towards a new state".

L75 "At the end of the 2000-year simulation with highest variability amplitude" SEP Along with these long-term, high-end numbers it would be instructive to give a perspective of what this would mean for a typical centennial timescale projection. Guessing from the figure, the drift in the calving ensembles is somewhere between 7 and 14 mm for a 100 year projection. If I understand well, the lower number (equal to 0.1 %) is discussed elsewhere as the criterium to define an acceptable ensemble range (l66). I think that could be compared and mentioned here.
We have added the drift amount for the first 100 years and also further context for these numbers in terms of sea level equivalent and comparison to ISMIP6 median projections. In the first 100 years, the drift in the highest simulation is about 1 cm SLE, compared to 10-20 cm median SLE contribution in ISMIP6. So, not negligible (and certainly not negligible when specific glaciers are considered, as we do later in this analysis).

L77 "without stochastic variability, but with a 270% increase in the mean calving rate". If the constant increase of the calving rate by 270% gives similar mass loss as the 1/3 mean stochastic case, what is the temporal structure of the amplitude variations in the forcing and in the simulated calving rate of the latter. Is it possible to visualise the two experiments alongside? What are the peak rates and how often are they coming out of the parametrisation and how realistic are these compared to e.g. observed speedup events for Sermeq Kujalleq? What is the time step of the stochastic variability? Is it comparable to observed (seasonal) speed changes?
As discussed in the text, the temporal structure of imposed calving variability is white noise (Gaussian, no autocorrelation) with stochastic perturbations added on a yearly time scale. It would be difficult to compare directly to observations of calving variability at Sermeq Kujalleq given the lack of long-term measurements of calving, as opposed to terminus variability which is driven by many processes. As argued in the above response, even just considering thermal forcing which is thought to drive a proportional response in frontal ablation, a standard deviation of 1/3 is quite reasonable (or even conservatively low) for most glaciers around Greenland. Thus, we consider the stochastic variability in this study to be of a realistic magnitude (in fact, probably a low estimate). As plotted above, the spatial pattern of thickness change in the 270% increase in calving flux simulation is very similar to the stochastic ensemble exhibiting noise-induced drift, though we do not feel much is added in including this figure in the main text. A discussion of this similarly has been added to the main text, however.

L80 "A deterministic model [...] would require tuning far away from true parameter values" I don't think true parameter values exists. For the type of models discussed here, tuning is always a calibration with compensating effects. Maybe "original parameter values"?
This sentence is perhaps more confusing than it needs to be. Changed to "Calibrating a deterministic model to match the observed ice sheet state, which is subject to variability from climatic and glaciological processes, would require tuning parameters to very different values."

L267 "start from a calibrated initial state" Not all of the models participating in ISMIP6 have a calibrated initial state. Reword to "e.g., with many of the models participating in the recent ISMIP6 intercomparisons".
Good point. Changed.

L270 "Other recent modeling studies use this same spin-up procedure" Adjust similarly in response to comment in L267. For example "Other recent modeling studies use a calibrated initial state, but then re-calibrate"

Changed with this wording

L277 "we recommend including internal variability in the forcing of ice sheet models, both during spin-up and transient simulations" I disagree with this recommendation, because it does not follow from the analysis in the paper. You have shown with your modelling that including internal variability after the spin-up can cause model drift. I think the recommendation that can be clearly derived from these results is that for projections one should avoid initialising a model to a deterministic forcing and then apply stochastic forcing. I think it would be great to have some recommendation about how to use stochastic forcing during the spin-up, but the paper currently provides no analysis of that case. See also my general point #2.

We answer this point at length in response to general point #2 above. The central point is that noise-induced drift and the tendency imposed by it are real and expected to occur in real ice sheets (since ice sheets are subject to stochastic forcing, and we know well from observations that they exhibit bifurcation and nonlinear dynamics). Ice sheet models which do not represent this aspect of the real system are thus not doing an effective job of representing reality, and the problem cannot be avoided simply by not forcing models with variability. We now provide two possible solutions to this problem in the revised text.